# Neural Methods for Logical Reasoning over Knowledge Graphs

**Alfonso Amayuelas**[*][†]**, Shuai Zhang**[†]**, Susie Xi Rao**[†] **& Ce Zhang**[†]
[*]EPFL
[†]ETH Zurich
`alfonso.amayuelas@alumni.epfl.ch`
`{shuazhang, raox, ce.zhang}@inf.ethz.ch`

## Abstract

Reasoning is a fundamental problem for computers and deeply studied in Artificial Intelligence. In this paper, we specifically focus on answering multi-hop logical queries on Knowledge Graphs (KGs). This is a complicated task because, in real-world scenarios, the graphs tend to be large and incomplete. Most previous works have been unable to create models that accept full First-Order Logical (FOL) queries, which include negative queries, and have only been able to process a limited set of query structures. Additionally, most methods present logic operators that can only perform the logical operation they are made for. We introduce a set of models that use Neural Networks to create one-point vector embeddings to answer the queries. The versatility of neural networks allows the framework to handle FOL queries with Conjunction ($\wedge$), Disjunction ($\vee$) and Negation ($\neg$) operators. We demonstrate experimentally the performance of our model through extensive experimentation on well-known benchmarking datasets. Besides having more versatile operators, the models achieve a 10% relative increase over the best performing state of the art and more than 30% over the original method based on single-point vector embeddings.

## 1 Introduction

Knowledge graphs (KGs) are a type of data structure that can capture many kinds of relationships between entities (e.g.: $Moscow \xrightarrow{\text{cityIn}} Russia$) and have been popularized since the creation of the semantic web or its introduction into Google's search engine. They can contain many kinds of different information, and they can be widely used in question-answering systems, search engines, and recommender systems (Palumbo et al., 2017; Xiong et al., 2017a).

Reasoning is a fundamental skill of human brains. For example, we can infer new knowledge based on known facts and logic rules, and discern patterns/relationships to make sense of seemingly unrelated information. It is a multidisciplinary topic and is being studied in psychology, neuroscience, and artificial intelligence (Fagin et al., 2003). The ability to reason about the relations between objects is central to generally intelligent behavior. We can define reasoning as the process of inferring new knowledge based on known facts and logic rules. Knowledge graphs are a structure used for storing many kinds of information, therefore the ability to answer complex queries and extract answers that are not directly encoded in the graph are of high interest to the AI community.

To answer complex queries, the model receives a query divided in logical statements. A full First-Order Logic (FOL) is necessary to process a wider range of queries, which includes negative queries. FOL includes the following logical operators: Existential ($\exists$), Conjunction ($\wedge$), Disjunction ($\vee$), and Negation ($\neg$). The power of representation of our logical framework is the key to process complex queries. However, most frameworks have only been able to process Existential Positive First-Order Logic (EPFO), which means that negative queries cannot be processed.

For example, One could ask a knowledge graph containing drugs and side effects the following question: *"What drug can be used to treat pneumonia and does not cause drowsiness?"*. The first step to answer such a query is to translate it into logical statements: $q = V_? \cdot \exists V : Treat(Pneumonia, V_?)$

---

Source code available on: https://github.com/amayuelas/NNKGReasoning

q = V$_?$ · ∃ V : *CitizenOf( Brazil, V )* ∧ *Winner(Ballon d'Or, V)* ∧ *Team(V, V$_?$)*

**(A) COMPUTATION GRAPH**

**(B) VECTOR SPACE**

Figure 1: **MLP Framework for KG Reasoning.** Representation of a sample query: "*List the teams where Brazilian football players who were awarded a Ballon d'Or played*". **(A)** Query represented by its logical statements and dependency graph. **(B)** 2D Representation of the answer entities in a one-point vector space used by the reasoning framework.

∧ ¬ *Cause(Drowsiness,V$_?$)*. Once the query is divided into logical statements, we obtain the computation graph, a directed acyclic graph (DAG) which defines the order of operations. Afterwards, we can start traversing the graph. However, many real-world graphs are incomplete and therefore traversing them becomes very hard and even computationally impossible. There are many possible answer entities, and it requires modeling sets of entities. As such, embedding methods become a good solution to answer these queries. Previous works (Hamilton et al., 2018; Ren et al., 2020; Ren & Leskovec, 2020) have created methods for embedding the query and the graph into a vector space. The idea of graph embeddings reduces the problem to simply using nearest-neighbor search to find the answers, without paying attention to the intermediate results.

The embedding approach solves many of the problems of query-answering in knowledge graphs. In theory, we could answer the queries just by traversing the graph. In practice, graphs are large and incomplete, and answering arbitrary logical queries becomes a complicated task. The graph incompleteness means that traversing its edges would not provide the correct answers.

This work aims to create some models that allow complex queries and extract the correct answers from large incomplete knowledge graphs. To this end, we present a set of models based on Neural Networks that embed the query and the entities into a one-point vector space. Then, it computes the distance between the query and the entities to rank the answers according to the likelihood to answer the query. We use the versatility of Neural Networks to create the operators needed to process FOL queries.

We conduct experiments using well-known datasets for KG Reasoning: FB15k, FB15-237, and NELL. The experiments show that our models can effectively answer FOL and provide a noticeable improvement when compared with the state-of-the-art baselines. Our models provide a relative improvement of 5% to 10% to the latest state-of-art method and about 30% to 40% when compared with the method that uses the same idea of one-point vector space embeddings (Hamilton et al., 2018).

The main contributions of this work are summarized as: (1). **New embedding-based methods for logical reasoning over knowledge graphs**: two new models, plus variants, for KG Reasoning. These methods embed the query and the entities in the same vector space with single-point vectors. Implementing the logical operators with neural networks provides versatility to create any operator with virtually the same architecture. (2). **Improved performance over the current state of the art**. Experimental results show that the models presented in this paper outperform the selected baselines: Graph Query Embeddings (GQE) (Hamilton et al., 2018), Query2Box (Q2B) (Ren et al., 2020), and BetaE (Ren & Leskovec, 2020). (3). **Handling of negative queries**. Modelling queries with negation has been an open question in KG Reasoning until recently. BetaE (Ren & Leskovec, 2020) introduced the first method able to do so. This work takes advantages of the good relationship inference capabilities of Neural Networks and uses them to create the negation operator.

## 2 RELATED WORK

Traditional tasks on graphs include *Link Prediction* (Liben-Nowell & Kleinberg, 2007), *Knowledge Base Completion* (Wang et al., 2015), or basic Query-Answering (one-hop). They are all different versions of the same problem: *Is link (h,r,t) in the KG?* or *Is t an answer to query (h,r,)?*, where only a variable is missing. However, we face a more complicated problem, known as Knowledge Graph Reasoning, that may involve several unobserved edges or nodes over massive and incomplete KGs. In this case, queries can be path queries, conjunctive queries, disjunctive queries or or a combination of them. A formal definition of KG Reasoning can be found in Chen et al. (2020), as stated in Definition 2.1.

**Definition 2.1** (Reasoning over knowledge graphs). Defining a knowledge graph as: $\mathcal{G} = \langle \mathcal{E}, \mathcal{R}, \mathcal{T} \rangle$, where $\mathcal{E}, \mathcal{T}$ represent the set of entities, $\mathcal{R}$ the set of relations, and the edges in $\mathcal{R}$ link two nodes to form a triple as $(h, r, t) \in \mathcal{T}$. Then, reasoning over a $KG$ is defined as creating a triplet that does not exist in the original $KG$, $\mathcal{G}' = \{(h, r, t) | h \in \mathcal{E}, r \in \mathcal{R}, t \in \mathcal{T}, (h, r, t) \notin \mathcal{G}\}$

Most related to our work are embedding approaches for multi-hop queries over KGs: (Hamilton et al., 2018), (Ren et al., 2020), (Ren & Leskovec, 2020) and (Das et al., 2016), as well as models for question answering (Yasunaga et al., 2021), (Feng et al., 2020). The main differences with these methods rely on the ability to handle full First-Order Logical Queries and using Neural Networks to define all logical operators, including the projection. We also deliver a more extensive range of network implementations.

On a broader outlook, we identify a series of works that aim to solve Knowledge Graph Reasoning with several different techniques, such as *Attention Mechanisms* (Wang et al., 2018), *Reinforcement Learning* like DeepPath (Xiong et al., 2017b) or DIVA (Chen et al., 2018), or *Neural Logic Networks* (Shi et al., 2020), (Qu & Tang, 2019).

## 3 MODELS

Both models presented here follow the idea behind Graph Query Embedding – GQE (Hamilton et al., 2018): Learning to embed the queries into a low dimensional space. Our models differ from it in the point that logical query operations are represented by geometric operators. In our case, we do not follow the direct geometric sense and these operators are all represented by Neural Networks, instead of just the Intersection operator in GQE. Similarly, however, the operators are jointly optimized with the node embeddings to find the optimal representation.

In order to answer a query, the system receives a query $q$, represented as a DAG, where the nodes are the entities and the edges the relationships. Starting with the embeddings $\mathbf{e}_{v_1}, ..., \mathbf{e}_{v_n}$ of its anchor nodes and apply the logical operations represented by the edges to finally obtain an embedding $\mathbf{q}$ of the query (Guu et al., 2015).

### 3.1 FORMAL PROBLEM DEFINITION

A Knowledge Graph ($\mathcal{G}$) is a heterogeneous graph with a set of entities – nodes – ($\mathcal{V}$) and a set of relations – edges – ($\mathcal{R}$). In heterogeneous graphs, there can be different kinds of relations, which are defined as binary functions $r : \mathcal{V} \times \mathcal{V} \rightarrow \{True, False\}$ that connect two entities with a directed edge. The goal is to answer First-Order Logical (FOL) Queries. We can define them as follows:

**Definition 3.1** (First-Order Logical Queries). A first-order logical query $q$ is formed by an anchor entity set $\mathcal{V}_a \subseteq \mathcal{V}$, an unknown target variable $V_?$ and a series of existentially quantified variables $V_1, ..., V_k$. In its disjunctive normal form (DNF), it is written as a disjunction of conjunctions:

$$q[V_?] = V_? \cdot \exists V_1, ...V_k : c_1 \vee c_2 \vee ... \vee c_n \tag{1}$$

where $c_i$ represents a conjunctive query of one or several literals: $c_i = \mathbf{e}_{i1} \wedge \mathbf{e}_{i2} \wedge ... \wedge \mathbf{e}_{im}$. And the literals represent a relation or its negation: $\mathbf{e}_{ij} = r(v_i, v_j)$ or $\neg v(v_i, v_j)$ where $v_i, v_j$ are entities and $r \in \mathcal{R}$.

The entity embeddings are initialized to zero and later *learned* as part of the training process, along with the operators' weights.

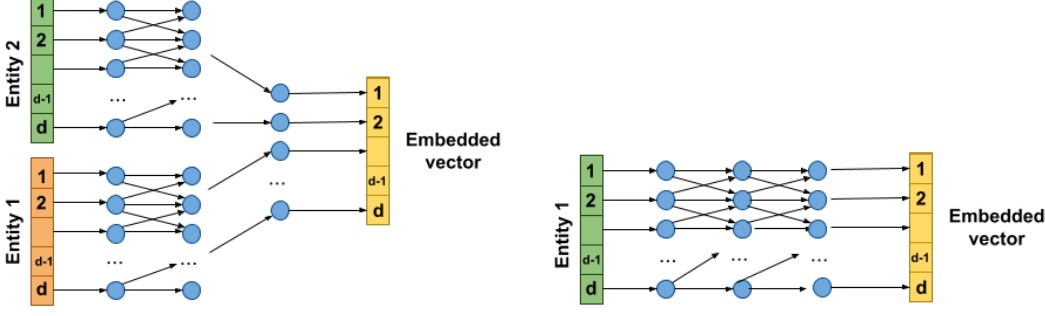

(a) Representation of MLP for 2 input operators: Projection, Intersection.

(b) Representation of MLP for 1 input operator: Negation.

Figure 2: Multi-Later Perceptron Model (MLP) - Network Architecture.

**Computation Graph**. The Computation Graph can be defined as the Direct Acyclic Graph (DAG) where the nodes correspond to embeddings and the edges represent the logical operations. The computation graph can be derived from a query by representing the relations as projections, intersections as merges and negation as complement. This graph shows the order of operations to answer the queries. Each branch can be computed independently and then merged until the sink node is reached. Each node represents a point in the embedding space and each edge represents a logical operation, computed via a Neural Network in our case. The representation of a FOL as a computation graph can be seen as a heterogeneous tree where each leaf node corresponds to the anchor entities and the root is the final target variable, which is a set of entities. The logical operations corresponding to the edges are defined below:

- *Projection*. Given an entity $v_i \in \mathcal{V}$ and a relation type $r \in \mathcal{R}$. It aims to return the set of adjacent entities with that relation. Being $P_r i(v_i, r)$ the set of adjacent entities through $r$, we define the projection as: $P_r i(v_i, r) = \{v' \in \mathcal{V} : (v, v') = \text{True}\}$.

- *Intersection*. The intersection can be defined as: $I(v_i) = \cap_{i=1}^{n} v_i$.

- *Negation*. It calculates the complement of a set of entities $\mathcal{T} \subseteq \mathcal{V}$: $N(\mathcal{T}) = \overline{\mathcal{T}} = \mathcal{V} \setminus \mathcal{T}$, where the set can either be the embedding corresponding to an entity or another embedding in between which represents a set of them.

A Union operation is unnecessary, as it will be later discussed in Sections 3.5. Query2Box (Ren et al., 2020) shows that a union operator becomes intractable in distance-based metrics.

### 3.2  MULTI-LAYER PERCEPTRON MODEL (MLP)

Based on the good results of Neural Tensor Networks (NTN) (Socher et al., 2013) for knowledge base completion, we have extended a similar approach to multi-hop reasoning.

We introduce three logical operators to compute the queries. Each of them is represented by a simple neural network: a multilayer perceptron. Each perceptron contains a feed-forward network: a linear layer plus a ReLu rectifier. The number of layers remains as a hyper-parameter. Figures 2a and 2b show what the model looks like.

**Neural operators.** We define the operators with a multi-layer perceptron. The model will take as an input the embedding representation of the input entities and will return an approximate embedding of the answer. Defining the operators with a Neural Network has the advantage of generalization. Thus, we distinguish between 2-input operators, Projection and Intersection and 1-input operator, Negation.

*- 2-input operator* (Figure 2a): Projection $P$, and Intersection $I$. The operator is composed by a multi-layer perceptron that takes 2 inputs and returns 1 as embedding as the output. The training process will make the networks learn the weights to represent each operation. Equation 2 expresses it formally:

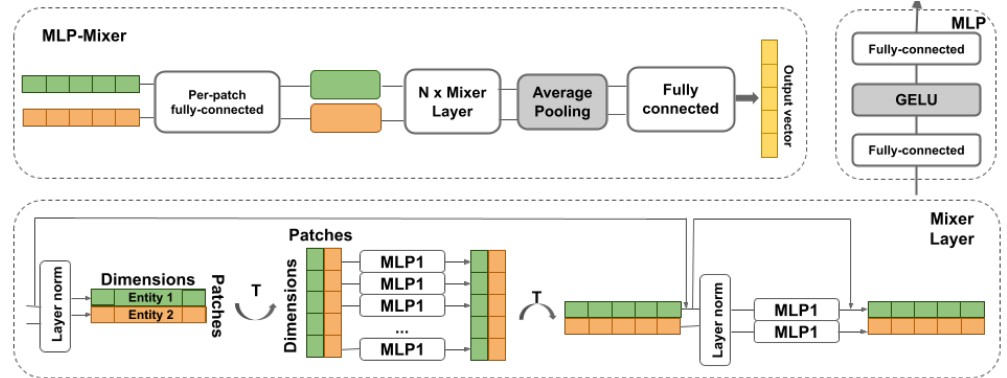

Figure 3: MLP-Mixer. At the top, we show the block diagram of the MLP-Mixer Architecture. It is formed by a per-patch fully connected module, N Mixer Modules, an average pooling and a last fully connected module. The bottom figure shows the Mixer Module, which contains one channel-mixing MLP, each consisting of 2 fully-connected layers and a ReLu nonlinearity. It also includes skip-connections, dropout and layer norm on the channels.

$$P(s_i, r_j) = NN_k(s_i, r_j), \forall s_i \in \mathcal{S}, \forall s_j \in \mathcal{R}$$
$$I(s_i, s_j) = NN_k(s_i, s_j), \forall s_i, s_j \in \mathcal{S}$$
$$(2)$$

where $s_i \in \mathcal{S}$ is an embedding in the vector space $\mathcal{S}$, $r_j \in \mathcal{R}$ is a relation and $NN_k$ is a multi-layer perceptron with $k$ layers.

The intersection can take more than two entities as an input, for instance the *3i* query structure. In this case we do a recursive operation, we use the result of the previous intersection to compute the next one.

- *1-input operator* (Figure 2b): Negation $N$. The goal of this operator is to represent the negation of a set of entities. Following the same neural network approach, we can represent it as in the equation below (Equation 3).

$$N(s_i) = NN_k(s_i), \forall s_i \in \mathcal{S} \qquad (3)$$

where $s_i \in \mathcal{S}$ is a vector in the embedding space, it can be an entity or the result of a previous operation. $NN_k$ is a multi-layer perceptron with $k$ layers and the same number of inputs as outputs.

### 3.3 MULTI-LAYER PERCEPTRON MIXER MODEL (MLP-MIXER)

The MLP-Mixer (Tolstikhin et al., 2021) is a Neural Architecture originally built for computer vision applications, which achieves competitive results when compared to Convolutional Neural Networks (CNNs) and Attention-based networks.

The MLP-Mixer is a model based exclusively on multilayer perceptrons (MLPs). It contains two types of layers: (1) one with MLPs applied independently to patches and (2) another one with MLPs applied across patches. Figure 3 presents a diagram of the architecture.

**Mixer operators.** We use the same procedure as in the MLP model. We use a full MLP-Mixer block to train each of the 2 operators with 2 inputs: projection and intersection. Since negation only has 1 input, the architecture cannot be accommodated for this use so far.

- *2-input operator* (Figure 2a). Represents Projection $P$ or Intersection $I$ with MLP-Mixer architecture.

$$P(s_i, r_j) = MLP_{mix}(s_i, r_j), \forall s_i \in \mathcal{S}, \forall s_j \in \mathcal{R}$$
$$I(s_i, s_j) = MLP_{mix}(s_i, s_j), \forall s_i, s_j \in \mathcal{S}$$
$$(4)$$

where $MLP_{mix}$ represent the mixer architecture, $s_i$ an embedding in the entity vector space $\mathcal{S}$; and $r_j \in \mathcal{R}$ a relation.

## 3.4 Training objective, Distance and Inference

**Training Objective**. The goal is to jointly train the logical operators and the node embeddings, which are learning parameters that are initialized randomly. Our training objective is to minimize the distance between the query and the query vector, while maximizing the distance from the query to incorrect random entities, which can be done via negative samples. Equation 5 expresses this training objective in mathematical terms.

$$\mathcal{L} = -\log \sigma(\gamma - \text{Dist}(v; q)) - \sum_{j=1}^{k} \frac{1}{k} \log \sigma(\text{Dist}(v_j'; q) - \gamma)) \tag{5}$$

where $q$ is the query, $v \in [q]$ is an answer of query (the positive sample); $v_j' \notin [q]$ represents a random negative sample and $\gamma$ refers to the margin. Both, the margin $\gamma$ and the number of negative samples $k$ remain as hyperparameters of the model.

**Distance measure**. When defining the training objective, we still need to specify the distance measure to compare the entity vectors. Unlike in previous works, we do not need a measure that compares between boxes or distributions. The Euclidean distance is enough for this purpose, as it calculates the distance between two points in a Euclidean space: $\text{Dist}(v, q) = |v - q|$.

**Inference**. Each operator provides an embedding. Following the query's computation graph, we obtain the final answer embedding (or query representation). Then, all entities are ranked according to the distance value of this embedding to all entity embeddings via near-neighbor search in constant time using Locality Sensitivity Hashing (Indyk & Motwani, 1998).

## 3.5 Discussion on answering FOL queries

The aim of this work is to answer a wide set of logical queries, specifically, be able to answer *first-order logical queries* (FOL), which includes: conjunctive $\wedge$, disjunctive $\vee$, existential $\exists$ and $\neg$ negation operations. Notice that we will not consider universal quantification ($\forall$) as it does not apply to real-world knowledge graphs since no entity will ever be connected to all other entities in the graph.

Theorem 1 in Query2Box shows that any embedding-based method that retrieves entities using a distance-based method is not able to handle arbitrary disjunctive queries. To overcome this problem, they transformed the queries into its Disjunctive Normal Form (DNF). By doing so, the disjunction is placed at the end of the computational graph and can be easily aggregated. The transformed computational graphs are equivalent to answering $N$ conjunctive queries. $N$ is meant to be small in practice, and all the $N$ computations can be parallelized. As expressed in (Davey & Priestley, 2002), all First-Order Logical Queries can be transformed into its DNF form. We refer readers to (Ren et al., 2020) to understand the transformation process.

## 4 Experiments and Results

### 4.1 Datasets

We perform experiments on three standard datasets in KG benchmarks. These are the same datasets used in Query2Box (Ren et al., 2020) and BetaE (Ren & Leskovec, 2020): FB15k (Bordes et al., 2013), FB15k-237 (Toutanova et al., 2015) and NELL995 (Xiong et al., 2017b).

In the experiments, we use the standard evaluation scheme for Knowledge Graphs, where edges are split into training, test and validation sets. After augmenting the KG to also include inverse relations and double the number of edges in the graph, we effectively create 3 graphs: $\mathcal{G}_{\text{train}}$ for training; $\mathcal{G}_{\text{valid}}$, which contains $\mathcal{G}_{\text{train}}$ plus the validation edges; and $\mathcal{G}_{\text{test}}$ which contains $\mathcal{G}_{\text{valid}}$ and the test edges. Some statistics about the datasets can be found in Appendix A.

### 4.2 Query/Answer generation

To obtain the queries from the datasets and its ground truth answers, we consider the 9 query basic structures from Query2box (Ren et al., 2020), shown in Figure 4. The training, validation, and test graphs were created as: $\mathcal{G}_{\text{train}} \subseteq \mathcal{G}_{\text{val}} \subseteq \mathcal{G}_{\text{test}}$, therefore the generated queries are also: $[\![q]\!]_{\text{train}} \subseteq$

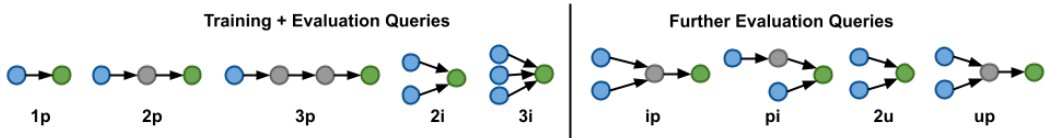

Figure 4: Training and evaluation queries represented with their graphical structures and abbreviation of their computation graphs. We consistently use the following nomenclature: *p* projection, *i* intersection, *n* negation, and *u* union.

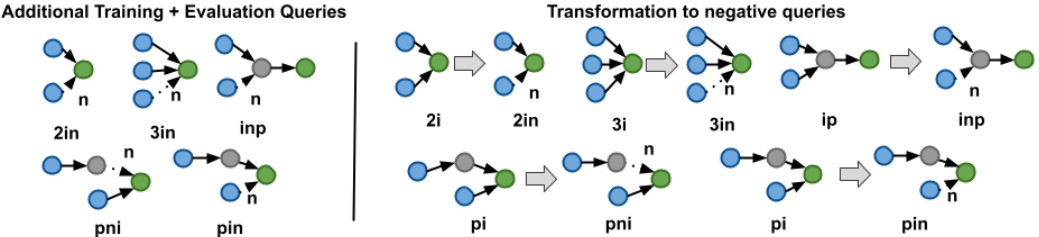

Figure 5: Queries with negation used in the experiment used for both, training and testing. On the right side, we show the transformation process from the original queries to its negative structure.

$[\![q]\!]_{\text{val}} \subseteq [\![q]\!]_{\text{test}}$. Thus, we evaluate and tune hyperparameters on $[\![q]\!]_{\text{val}} \setminus [\![q]\!]_{\text{train}}$ and report the results on $[\![q]\!]_{\text{test}} \setminus [\![q]\!]_{\text{val}}$. We always evaluate on queries and entities that were not part of the already seen dataset used before.

For the experiments, we have used the train/valid/test set of queries-answers used for training and evaluating BetaE (Ren & Leskovec, 2020). This query-generation system differs from the original in the fact that it limits the number of possible answers to a specific threshold since some queries in Query2box had above 5000 answers, which is unrealistic. More information about how queries are created in BetaE can be found in Appendix A.

To include FOL queries, BetaE (Ren & Leskovec, 2020) created some transformations to include 5 additional query structures with negation, shown in the right panel of Figure 5.

### 4.3 EVALUATION

Given the rank of answer entities $v_i \in \mathcal{V}$ to a non-trivial test query $q$, we compute the evaluation metrics defined according to Equation 6 below. Then, we average all queries with the same query format.

$$\text{Metric}(q) = \frac{1}{|[\![q]\!]_{\text{test}} \setminus [\![q]\!]_{\text{val}}|} \sum_{v_i \in \mathcal{V}} f_{\text{metric}}(\text{rank}(v_i)) \tag{6}$$

where $v_i$ is the set of answers $\mathcal{V} \subset [\![q]\!]_{\text{test}} \setminus [\![q]\!]_{\text{val}}$, $f_{\text{metric}}$ the specific metric function and $\text{rank}(v_i)$ the rank of answer entities returned by the model.

**Mean Reciprocal Rank (MRR)**: It is a statistic measure used in Information Retrieval to evaluate the systems that returns a list of possible responses ranked by their probability to be correct. Given an answer, this measure is defined as the inverse of the rank for the first correct answer, averaged over the sample of queries $\mathcal{Q}$. Equation 7 express this measure formally.

$$MRR = \frac{1}{|\mathcal{Q}|} \sum_{i=1}^{|\mathcal{Q}|} \frac{1}{\text{rank}_i} \tag{7}$$

where $\text{rank}_i$ refers to the rank position of the first correct/relevant answer for the i-th query.

**Hits rate at K (H@K)**: This measure considers how many correct answers are ranked above K. It directly provides an idea of how the system is performing. Equation 8 defines this metric mathematically.

$$H@K = \mathbf{1}_{v_i \leq K} \tag{8}$$

## 4.4 BASELINES AND MODEL VARIANTS

To compare our results, we have selected the state-of-the-art baselines from some previous papers on KG Logical Reasoning: Graph Query Embeddings (Hamilton et al., 2018), Query2Box (Ren et al., 2020) and BetaE (Ren & Leskovec, 2020). Only the last method, BetaE, accepts negative queries.

We have modified the simpler model, MLP, to potentially improve the initial results. Below, we describe the 3 modifications:

**Heterogeneous Hyper-Graph Embeddings** (Sun et al., 2021). The Heterogeneous Hypergraph Embeddings creates graph embeddings by projecting the graph into a series of snapshots and taking the Wavelet basis to perform localized convolutions. In essence, it is an embedding transformation that aims to capture the information of related nodes. We have added this transformation to our MLP model right before computing the distance, and then we calculate the distance measure defined specifically for this new hyper space.

**Attention mechanism** (Vaswani et al., 2017). The goal of attention layers is to enhance the "important" parts of the input data and fade out the rest. This allows the modelling of dependence without regard to the length of the input. This method has proven a performance increase in many Deep Learning Applications. We have implemented an attention mechanism for the intersection operator in the MLP model.

**2-Vector Average Approach**. In our models, we create an embedding of the query and the graph entities to a point in a hyperdimensional space. Since we are using Neural Networks to create these embeddings, the resulting embeddings will depend on the training process and the optimization of the weights. We have no assurance the embedding is correct. To add more robustness to the computation, we have decided to calculate the embedding twice with two separate networks and average the results between the two networks.

## 5 RESULTS

Table 1 shows the results of MRR for our models – MLP and MLP-Mixer – when compared to the baselines – GQE, Q2B, BetaE –. As listed in the table, our methods yield promising results that improve those from the state of the art across all query forms and datasets. Additionally, Table 2 shows the results for the model variants previously described in Section 4.4. We observe the Hyper Embedding space does not provide an improvement when compared to the basic version of the model. On the other side, the other two variants do show a significance improvement. Nonetheless, it is worth noting the 2-vector approach may improve the result at the cost of more computer power, as it mainly computes each embedding twice. Finally, in Table 3 we also show results for experiments on full FOL queries, which include negation. Additional results for H@1 can be found in Appendix C.

### 5.1 ANALYSIS OF RESULTS

In general terms, we observe an increased performance over the selected baselines. In this section we discuss some of the implications:

- *Implementation of Logical Operators*. One of the main differences with other approaches such as GQE (Hamilton et al., 2018) or Q2B (Ren et al., 2020) is the use of Neural Networks to represent all logical operations: Projection, Intersection and Negation. In comparison to GQE that only uses it on the Intersection, or Q2B that creates a set of geometrical operations to represent the logical operations. In light of these results, it seems that the geometrical implications of the operations can be restraining some of the possible solutions.

- *Correctly learning the logical operations*. It is hard to clearly find out if the operators are learning the logical operations correctly. At least, we can assure the Neural Networks do a better job at learning the approximate solution. Neural Networks that implement directly a logical operation are already currently under research (Shi et al., 2019). A too constrained implementation of a Neural Logic Network can be found in Appendix D.

- *Performance of model variants.* We observe that both Attention Mechanism and the 2-Vector Approach manage to improve the results from the original MLP model. This indicates that Hyper-Graph Embeddings are not correctly transforming the embedding space in our case. Additionally, the improvement original from the 2-vector approach seems to indicate the optimal solution was not yet reached and there is still room for improvement on learning the correct logical operations.

Table 1: MRR Results (%) of baselines (GQE, Q2B, BetaE) and our models (MLP, MLP-Mixer) on EPFO ($\exists, \wedge, \vee$) queries.

| Dataset | Model | 1p | 2p | 3p | 2i | 3i | ip | pi | 2u | up | avg |
|---|---|---|---|---|---|---|---|---|---|---|---|
| | **MLPMix** | **69.7** | 27.7 | 23.9 | **58.7** | **69.9** | 30.8 | **46.7** | **38.2** | 24.8 | 43.4 |
| | **MLP** | 67.1 | **31.2** | **27.2** | 57.1 | 66.9 | **33.9** | 45.7 | 38.0 | **28.0** | **43.9** |
| FB15k | **BetaE** | 65.1 | 25.7 | 24.7 | 55.8 | 66.5 | 28.1 | 43.9 | 40.1 | 25.2 | 41.6 |
| | **Q2B** | 68.0 | 21.0 | 14.2 | 55.1 | 66.5 | 26.1 | 39.4 | 35.1 | 16.7 | 38.0 |
| | **GQE** | 54.6 | 15.3 | 10.8 | 39.7 | 51.4 | 19.1 | 27.6 | 22.1 | 11.6 | 28.0 |
| | **MLPMix** | 42.4 | 11.5 | 9.9 | **33.5** | **46.8** | 14.0 | **25.4** | **14.0** | 9.2 | **22.9** |
| | **MLP** | **42.7** | **12.4** | **10.6** | 31.7 | 43.9 | **14.9** | 24.2 | 13.7 | **9.7** | 22.6 |
| FB15k-237 | **BetaE** | 39.0 | 10.9 | 10.0 | 28.8 | 42.5 | 12.6 | 22.4 | 12.4 | **9.7** | 20.9 |
| | **Q2B** | 40.6 | 9.4 | 6.8 | 29.5 | 42.3 | 12.6 | 21.2 | 11.3 | 7.6 | 20.1 |
| | **GQE** | 35.0 | 7.2 | 5.3 | 23.3 | 34.6 | 10.7 | 16.5 | 8.2 | 5.7 | 16.3 |
| | **MLPMix** | 55.4 | 16.5 | 13.9 | **39.5** | **51.0** | 18.3 | **25.7** | 14.7 | 11.2 | **27.4** |
| | **MLP** | 55.2 | **16.8** | **14.9** | 36.4 | 48.0 | 18.2 | 22.7 | **14.7** | **11.3** | 26.5 |
| NELL995 | **BetaE** | 53.0 | 13.0 | 11.4 | 37.6 | 47.5 | 14.3 | 24.1 | 12.2 | 8.5 | 24.6 |
| | **Q2B** | 42.2 | 14.0 | 11.2 | 33.3 | 44.5 | 16.8 | 22.4 | 11.3 | 10.3 | 22.9 |
| | **GQE** | 32.8 | 11.9 | 9.6 | 27.5 | 35.2 | 14.4 | 18.4 | 8.5 | 8.8 | 18.6 |

Table 2: MRR Results (%) of the model variants of MLP on EPFO ($\exists, \wedge, \vee$) queries: Hyper Embedding space, Attention mechanism and 2-vector approach.

| Dataset | Model | 1p | 2p | 3p | 2i | 3i | ip | pi | 2u | up | avg |
|---|---|---|---|---|---|---|---|---|---|---|---|
| | **2-vector** | **71.9** | **32.1** | **27.1** | **59.9** | **70.5** | **33.7** | **48.4** | **40.4** | **28.4** | **45.8** |
| FB15k | **Attention** | 70.0 | 29.5 | 25.4 | 58.6 | 70.0 | 29.8 | 47.2 | 36.8 | 26.5 | 43.8 |
| | **HyperE** | 64.2 | 23.2 | 20.1 | 50.7 | 63.4 | 19.6 | 39.2 | 29.6 | 20.6 | 36.8 |
| | **2-vector** | **43.4** | **12.6** | **10.4** | **33.6** | **47.0** | **14.9** | **25.7** | **14.2** | **10.2** | **23.6** |
| FB15k-237 | **Attention** | 42.7 | 11.9 | 10.2 | 33.3 | 46.7 | 14.2 | 25.2 | 14.1 | 9.7 | 23.1 |
| | **HyperE** | 41.1 | 10.6 | 9.1 | 28.5 | 41.6 | 11.0 | 21.8 | 13.1 | 8.8 | 20.6 |
| | **2-vector** | **55.6** | **16.3** | **14.9** | **38.5** | **49.5** | 17.1 | **23.7** | 14.6 | **11.0** | **26.8** |
| NELL995 | **Attention** | **55.6** | 16.2 | 14.4 | 38.0 | 49.0 | **17.9** | 22.3 | **14.7** | **11.0** | 26.6 |
| | **HyperE** | 54.6 | 14.5 | 12.1 | 34.6 | 45.8 | 13.9 | 21.7 | 12.3 | 9.1 | 24.3 |

Table 3: MRR Results (%) of MLP model on full FOL queries ($\exists, \vee, \wedge, \neg$), including negative queries. We only use show results for negative query structures.

| Dataset | Model | 2in | 3in | inp | pin | pni | avg |
|---|---|---|---|---|---|---|---|
| FB15K | **MLP** | **17.2** | **17.8** | **13.5** | **9.1** | **15.2** | **14.5** |
| | **BetaE** | 14.3 | 14.7 | 11.5 | 6.5 | 12.4 | 11.8 |
| FB15-237 | **MLP** | **6.6** | **10.7** | **8.1** | **4.7** | **4.4** | **6.9** |
| | **BetaE** | 5.1 | 7.9 | 7.4 | 3.6 | 3.4 | 5.4 |
| NELL995 | **MLP** | **5.1** | **8.0** | **10.0** | **3.6** | **3.6** | **6.1** |
| | **BetaE** | **5.1** | 7.8 | **10.0** | 3.1 | 3.5 | 5.9 |

## 6 CONCLUSIONS

In this work, we have introduced a competitive embedding framework for Logical Reasoning over Knowledge Graphs. It presents a flexible approach to build logical operators through Neural Networks. This method accepts queries in its First-Order Logical form, and it is one of the first models to accept negative queries. Extensive experimental results show a significance performance improvement when compared to other state-of-the-art methods built for this purpose.

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

## A  QUERY GENERATION AND STATISTICS

Statistics about the datasets can be found in Table 4 However, it is useful to understand how they have been created. Given the 3 datasets, 3 graphs are created: $G_{\text{train}}$, $G_{\text{val}}$, $G_{\text{test}}$ for training, testing and validation, respectively. As explained in Section 4.2, $G_{\text{train}}$ contains the training edges, while $G_{\text{val}}$ contains edges from training + valid and $G_{\text{test}}$ from valid + test. Then they apply pre-order traversal to the target nodes until all anchor entities are instantiated, and use post-order traversal to find the answers. The difference with the query generation process used in Query2Box is a limit on the number of queries, since the original dataset had queries with over 5000 answers, nearly 1/3 of some datasets. The average number of queries and answers are shown in Tables 5 and 6.

Table 4: Datasets statistics.

| Dataset | Entities | Relations | Training Edges | Val Edges | Test Edges | Total Edges |
|---|---|---|---|---|---|---|
| FB15k | 14,951 | 1,345 | 483,142 | 50,000 | 59,071 | 592,213 |
| FB15k-237 | 14,505 | 237 | 272,115 | 17,526 | 20,438 | 310,079 |
| NELL995 | 63,361 | 200 | 114,213 | 14,324 | 14,267 | 142,804 |

Table 5: Number of queries divided by training, validation and test sets, and query structure.

| Queries | Training | | Validation | | Test | |
|---|---|---|---|---|---|---|
| Dataset | 1p/2p/3p/2i/3i | 2in/3in/inp/pin/pni | 1p | Rest | 1p | Rest |
| FB15k | 273,710 | 27,371 | 59,097 | 8,000 | 67,016 | 8,000 |
| FB15k-237 | 149,689 | 23,714,968 | 20,101 | 5,000 | 22,812 | 5,000 |
| NELL995 | 107,982 | 10,798 | 16,927 | 4,000 | 17,034 | 4,000 |

Table 6: Average number of answers divided by query structure.

| Dataset | 1p | 2p | 3p | 2i | 3i | ip | pi | 2u | up | 2in | 3in | inp | pin | pni |
|---|---|---|---|---|---|---|---|---|---|---|---|---|---|---|
| FB15k | 1.7 | 19.6 | 24.4 | 8.0 | 5.2 | 18.3 | 12.5 | 18.9 | 23.8 | 15.9 | 14.6 | 19.8 | 21.6 | 16.9 |
| FB15k-237 | 1.7 | 17.3 | 24.3 | 6.9 | 4.5 | 17.7 | 10.4 | 19.6 | 24.3 | 16.3 | 13.4 | 19.5 | 21.7 | 18.2 |
| NELL995 | 1.6 | 14.9 | 17.5 | 5.7 | 6.0 | 17.4 | 11.9 | 14.9 | 19.0 | 12.9 | 11.1 | 12.9 | 16.0 | 13.0 |

## B  EXPERIMENTAL DETAILS

Our code is implemented using PyTorch. For the baselines, we have used the implementation of the baselines and the testing framework from BetaE (Ren & Leskovec, 2020), this also includes the code for Query2Box (Ren et al., 2020) and GQE (Hamilton et al., 2018). For a fair comparison with the models in the paper, we have selected the same hyperparameters listed in the paper. This also includes the consideration of the single-point vector methods – GQE, MLP, MLPMixer – having embedding dimensions $2n$. Since Query2Box needs to model the offset and the center of the box and BetaE has 2 parameters per distribution: $\alpha$ and $\beta$, and the embedding dimensions is set to $n$.

All our models and GQE use the following parameters: Embed dim = 800, learning rate = 0.0001, negative sample size = 128, batch size = 512, margin = 24, num. iterations = 300,000/450,000. Q2B and BetaE differ from the previous configuration in Embed. dim = 400 and margin = 30/60.

All experiments have been computed on independent processes on NVIDIA GPUs, either the GeForce GTX Titan X Pascal (12 GB) or the Tesla T4 (16 GB).

# C  Additional results

*Hits at K* (H@K) is another metric of evaluation that captures the sense of precision of our model to retrieve the correct entities. It is described in the Section 4.3. In Table 7 we show the results of H@1 for all models (MLP-Mixer, MLP) and baselines (BetaE, Q2B, GQE). In Table 8 we show the H@1 results for our model variants: HyperE, Attention Layers and 2-vector average. Finally, Table 9 shows the results of BetaE and MLP on negative queries.

Table 7: H@1 Results (%) of baselines (GQE, Q2B, BetaE) and our models (MLP, MLP-Mixer) on EPFO ($\exists, \wedge, \vee$) queries.

| Dataset | Model | 1p | 2p | 3p | 2i | 3i | ip | pi | 2u | up | avg |
|---|---|---|---|---|---|---|---|---|---|---|---|
| FB15k | MLPMix | **59.0** | 18.6 | 16.2 | **47.5** | **59.8** | 21.3 | **35.8** | 26.8 | 16.3 | 33.5 |
| | MLP | 56.0 | **22.0** | **19.2** | 46.3 | 56.4 | **24.0** | 35.1 | **26.9** | **19.1** | **33.9** |
| | BetaE | 52.0 | 17.0 | 16.9 | 43.5 | 55.3 | 19.3 | 32.3 | 28.1 | 16.9 | 31.3 |
| | Q2B | 52.0 | 12.7 | 7.8 | 40.5 | 53.4 | 16.7 | 26.7 | 22.0 | 9.4 | 26.8 |
| | GQE | 34.2 | 8.3 | 5.0 | 23.8 | 34.9 | 11.2 | 15.5 | 11.5 | 5.6 | 16.6 |
| FB15k-237 | MLPMix | 31.9 | 6.0 | 4.7 | **22.7** | **36.5** | 8.2 | **16.7** | **7.7** | 4.3 | **15.4** |
| | MLP | **32.5** | **6.4** | **5.3** | 21.4 | 33.4 | **8.9** | 16.0 | 7.5 | 4.3 | 15.1 |
| | BetaE | 28.9 | 5.5 | 4.9 | 18.3 | 31.7 | 6.7 | 14.0 | 6.3 | **4.6** | 13.4 |
| | Q2B | 28.3 | 4.1 | 3.0 | 17.5 | 29.5 | 7.1 | 12.3 | 5.2 | 3.3 | 12.3 |
| | GQE | 22.4 | 2.8 | 2.1 | 11.7 | 20.9 | 5.7 | 8.4 | 3.3 | 2.1 | 8.8 |
| NELL995 | MLPMix | 45.3 | 10.8 | 9.1 | **28.5** | **40.0** | 12.1 | **18.2** | 8.4 | 6.3 | **19.8** |
| | MLP | **45.6** | **11.2** | **10.0** | 25.3 | 36.7 | **12.4** | 15.4 | **8.6** | **6.5** | 19.0 |
| | BetaE | 43.5 | 8.1 | 7.0 | 27.1 | 36.5 | 9.3 | 17.4 | 6.9 | 4.7 | 17.8 |
| | Q2B | 23.8 | 8.7 | 6.9 | 20.3 | 31.5 | 10.7 | 14.2 | 5.0 | 6.0 | 14.1 |
| | GQE | 15.4 | 6.7 | 5.0 | 14.3 | 20.4 | 9.0 | 10.6 | 2.9 | 5.0 | 9.9 |

Table 8: H@1 Results (%) of model variants: HyperEmbedddings, Attention Mechanism and 2-vector average.

| Dataset | Model | 1p | 2p | 3p | 2i | 3i | ip | pi | 2u | up | avg |
|---|---|---|---|---|---|---|---|---|---|---|---|
| FB15k | 2-vector | **71.9** | **32.1** | **27.1** | **59.9** | **70.5** | **33.7** | **48.4** | **40.4** | **28.4** | **45.8** |
| | Attention | 70.0 | 29.5 | 25.4 | 58.6 | 70.0 | 29.8 | 47.2 | 36.8 | 26.5 | 43.8 |
| | HyperE | 64.2 | 23.2 | 20.1 | 50.7 | 63.4 | 19.6 | 39.2 | 29.6 | 20.6 | 36.8 |
| FB15k-237 | 2-vector | **43.4** | **12.6** | **10.4** | **33.6** | **47.0** | **14.9** | **25.7** | **14.2** | **10.2** | **23.6** |
| | Attetion | 42.7 | 11.9 | 10.2 | 33.3 | 46.7 | 14.2 | 25.2 | 14.1 | 9.7 | 23.1 |
| | HyperE | 41.1 | 10.6 | 9.1 | 28.5 | 41.6 | 11.0 | 21.8 | 13.1 | 8.8 | 20.6 |
| NELL995 | 2-vector | **55.6** | **16.3** | **14.9** | **38.5** | **49.5** | 17.1 | **23.7** | 14.6 | **11.0** | **26.8** |
| | Attention | **55.6** | 16.2 | 14.4 | 38.0 | 49.0 | **17.9** | 22.3 | **14.7** | **11.0** | 26.6 |
| | HyperE | 54.6 | 14.5 | 12.1 | 34.6 | 45.8 | 13.9 | 21.7 | 12.3 | 9.1 | 24.3 |

Table 9: H@1 Results (%) of baselines – BetaE – and our model – MLP – on negative queries.

| Dataset | Model | 2in | 3in | inp | pin | pni | avg |
|---|---|---|---|---|---|---|---|
| FB15K | MLP | **8.3** | **8.6** | **6.9** | **3.6** | **7.4** | **6.9** |
| | BetaE | 6.4 | 6.6 | 5.5 | 2.2 | 5.2 | 5.2 |
| FB15-237 | MLP | **2.2** | **4.2** | **3.4** | **1.4** | **1.2** | **2.5** |
| | BetaE | 1.5 | 2.8 | 2.8 | 0.7 | 0.9 | 1.7 |
| NELL995 | MLP | 1.4 | **2.6** | 4.2 | **0.9** | **1.1** | 2.0 |
| | BetaE | **1.6** | **2.6** | **4.4** | **0.9** | **1.1** | **2.1** |

# D   APPENDIX: ADDITIONAL MODELS

Table 10: MRR Results (%) of additional models: CNN, NLN on EPFO ($\exists, \wedge, \vee$) queries.

| Dataset | Model | 1p | 2p | 3p | 2i | 3i | ip | pi | 2u | up | avg |
|---------|-------|------|------|-----|------|------|------|------|------|-----|------|
| **FB15k-237** | **CNN** | 41.1 | 11.1 | 9.6 | 29.6 | 41.4 | 11.9 | 21.4 | 13.0 | 9.3 | 20.9 |
| | **NLN** | 9.6 | 2.4 | 2.6 | 5.5 | 7.8 | 1.0 | 4.4 | 0.8 | 1.9 | 0.4 |

During this research, we have additionally explored other models, like Convolutional Neural Networks and Neural Logic Networks. We show the preliminary results of these models in the appendix for informative reasons, available in Table 10. They have been tested on FB15k-237.

**Convolutional Neural Networks (CNNs)** CNNs manage to capture the intrinsic dependencies that happen between close items. This is the reason why they are well suited for images, where closed pixels tend to have similar colors. This is not the case we find in Knowledge Graph embeddings, and we can intuitively its poor performance.

We have adapted a simple model of a CNN to compute the logic operators in our method. As previously, we have created two different CNNs, one with 2 inputs (Intersection and Projection) and another one with 1 input (Negation).

*- 1-input operator.* Represented by CNN with 2 convolutional layers (1st: *in_channels = 1, out_channels = 10, kernel_size = 6*, 2nd: *in_channels = 10, out_channels = 10, kernel_size = 6*) + maxPool (*kernel_size = 6*), followed by 3 fully connected layers + ReLu, except for the last one which does not have ReLu. Input and output sizes are the same.

*- 2-input operator.* It uses the same architecture as the 1-input operator, but applies the layers Conv+MaxPool to the 2 input entities separately and concatenates their outputs to feed the 3 fully connected layers.

**Neural Logic Networks (NLN)** Neural Logic Networks (Shi et al., 2019) are a kind of network architecture created to conduct logical inference. They use vectors to represent the variables, and each logic operation is learned as a neural model with some predefined logic regularizers. The logic regularizers constraint the neural module to complete the tasks they are conceived for. They have defined the required regularizers for the most common operations, like NOT, AND, and OR.

In this model, we have implemented the Intersection operator and used the same Projection operator from GQE (Hamilton et al., 2018): $P(q,r) = R_r q$, where $R_r^{d \times d}$ is a trainable parameter for edge type $r$ and $q$ an entity.

*- Intersection operator* The intersection operator is implemented as the AND logical operation. Below, we show the formal definition of the AND (Equation 9) operation – a basic multilayer perceptron – and its corresponding regularizer in Table 11.

$$I(v_i, v_j) = \text{AND}(v_i, v_j) = \mathbf{H}_{a2} f(\mathbf{H}_{a1}(v_i|v_j) + b_a) \tag{9}$$

where $\mathbf{H}_{a1} \in R^{d \times 2d}$, $\mathbf{H}_{a2} \in R^{d \times d}$ and $b_a \in R^d$ are the parameters of the Neural Network.

Table 11: Logical regularizer for the AND operation. $Sim(\cdot)$ is a measure of the similarity between two items, Euclidean distance in our case.

| Operation | Logic Rule | Equation | Logic Regularizer |
|-----------|-----------|----------|-------------------|
| | Identity | $w \wedge T = w$ | $r_1 = \sum_{w \in W} 1 - Sim(\mathbf{AND(w, T), w})$ |
| AND | Annihilator | $w \wedge F = F$ | $r_2 = \sum_{w \in W} 1 - Sim(\mathbf{AND(w, F), F})$ |
| | Idempotence | $w \wedge w = w$ | $r_3 = \sum_{w \in W} 1 - Sim(\mathbf{AND(w, w), w})$ |
| | Complementation | $w \wedge \neg w = F$ | $r_4 = \sum_{w \in W} 1 - Sim(\mathbf{AND(w, NOT(w)), F})$ |

*Comment.* Neural Logic Networks could be a good solution for Reasoning in KGs, especially after they have already been proven to be useful in other reasoning tasks (Shi et al., 2020). There are many reasons why it might have not worked in this case: a wrong implementation, a unit mismatch between our loss and the regularizer values, or the regularizer being too constraining for the task.

