# OpenReview forum: "Neural Methods for Logical Reasoning over Knowledge Graphs"
_ICLR.cc/2022/Conference — ICLR 2022 Poster_

### Official Review · Reviewer_zRD3 · 2021-10-26

**Correctness:** 3
**Technical Novelty And Significance:** 3
**Empirical Novelty And Significance:** 3
**Recommendation:** 8
**Confidence:** 3

**Main Review:**

The results obtained are promising and the possibility of managing negation is a plus that few systems can boast of having. The experiments conducted are based on standard datasets, so the comparison with competitors is significant. I found the paper to be mathematically correct and formal enough. The description of the system is sufficiently detailed.
A shortcoming of the paper is the related work section, which presents a list of related systems without significantly discussing the differences, their pros and cons compared to the system presented.


As minor issues, on page one, in the sentence 'directed acyclic graph (DAG) graph which defines ...' the word graph is repeated. While on page 6, sigma is defined as margin, but I think that the margin is gamma while sigma could be the sigmoid function. Is it right?

**Summary Of The Paper:**

The article presents a new system for solving multi-hop queries on knowledge graphs. The system allows queries to be performed by handling operators using one- and two-input MLPs. This formulation also makes it possible to handle negative queries.
The paper also presents a second version of the presented system that exploits an MLP Mixer model, usually used in computer vision.
The system was tested on three standard knowledge graphs: FB15K, FB15k-237 and NELL995. The results obtained are promising because the presented system is able to improve on the established baseline. The results obtained by the BetaE, Q2B and GQE systems on the same datasets were used as the baseline.

**Summary Of The Review:**

The results obtained are promising and the possibility of solving negative queries is a strong point in favour.
An appropriate discussion of related work is lacking.

---

> ### Author Response · Authors · 2021-11-22
> **As identified in your comment, a shortcoming of the paper was the Related Work section. We have modified this section to include more updated works and clearly state the differences to the most related works. Additionally, we have also corrected your described minor issues.**
>
> **Related Work**
>
> *A shortcoming of the paper is the related work section, which presents a list of related systems without significantly discussing the differences, their pros and cons compared to the system presented.*
>
> We have addressed your concerns on the related work and updated this section in the new version of the paper. Let us include the new related section to this comment, where we aim to provide more updated references and a comparison with the most related works.
>
> Traditional tasks on graphs include Link Prediction (Liben-Nowell & Kleinberg, 2007), Knowledge Base Completion(Wang et al., 2015), or basic Query-Answering (one-hop). They are all different versions of the same problem: *Is link (h,r,t) in the KG?* *Or Is t an answer to query (h,r,)?*, where only a variable is missing. However, we face a more complicated problem, known as Knowledge Graph Reasoning, that may involve several unobserved edges or nodes over massive and incomplete KGs. In this case, queries can be path queries, conjunctive queries, disjunctive queries or a combination of them. A formal definition of KG Reasoning can be found in Chen et al. (2020), as stated in Definition 2.1
>
> Definition 2.1 : *Reasoning over knowledge graphs*.
> Defining a knowledge graph as: $G=\langle E,R,T \rangle$, where $E,T$ represent the set of entities, $R$ the set of relations, and the edges in $R$ link two nodes to form a triple as $(h,r,t) \in T$. Then, reasoning over a $KG$ is defined as creating a triplet that does not exist in the original $KG$, $G'=\{(h,r,t)|h \in E, r \in R, t \in T, (h,r,t) \not\in G\}$
>
> Most related to our work are embedding approaches for multi-hop queries over KGs: (Hamiltonet al., 2018), (Ren et al., 2020), (Ren & Leskovec, 2020) and (Das et al., 2016), as well as models for question answering (Yasunaga et al., 2021), (Feng et al., 2020). The main differences with these methods rely on the ability to handle full First-Order Logical Queries and using Neural Networksto define all logical operators, including the projection. We also deliver a more extensive range of networks implementations.
>
> On a larger outlook, we identify a series of works that aim to solve Knowledge Graph Reasoning with several different techniques, such as *Attention Mechanisms* (Wang et al., 2018), *Reinforcement Learning *like DeepPath (Xiong et al., 2017b) or DIVA (Chen et al., 2018), or *Neural Logic Networks* (Shi et al., 2020), (Qu & Tang, 2019).

---

### Official Review · Reviewer_vpE2 · 2021-11-01

**Correctness:** 4
**Technical Novelty And Significance:** 3
**Empirical Novelty And Significance:** 2
**Recommendation:** 5
**Confidence:** 3

**Main Review:**

*Strengths*:
 + Good empirical performance
 + Good amount of information to reproduce the method given in the main text and appendix
 + Able to deal with negative queries

*Cons*:
 - Poor related work section: Most of the works cited are rather old (e.g. in KGR with logic rules the most recent is 2018, which in machine learning terms is old), there is no comparison of the method proposed and the baselines used, and classic KGR methods that answer queries of type (h,r,?) and complex queries should be clearly distinguished. The related work fails to explain the relation of this work to prior work in a meaningful way.
 - Clarity, the paper should outline in more detail the Graph Query Embedding method that this paper is based on.

*Questions*:
 1. Given more than 2 inputs to the intersection operator, how is the order of recursion decided? A different order will yield a different answer.
 2. What do the colour of the nodes mean in Figure 1?
 3. How is the subgraph (e.g. Figure 1 (C)) extracted from the Knowledge graph?
 4. How are the initial embeddings s_i for each entity and relation learned?
 5. How is the training objective dealing with several possible answers? (I think I know, but the main text should state it clearly)

*Minor*:
 - use \textit{NN} in mathmode to avoid the weird spacing between the two Ns

**Summary Of The Paper:**

The paper proposes to learn the projection, intersection, negation operators necessary for dealing with complex queries with neural networks. The neural networks are built on top of MLP-Mixer architecture and produce good empirical performance.

**Summary Of The Review:**

The paper's empirical performance is solid, the technical novelty lies in making the Graph Query Embedding method work with Neural Networks operators. However, the paper lacks clarity and the related work is poor. Happy to increase my score, should my concerns be addressed.

---

> ### Author Response · Authors · 2021-11-22
> **The paper has been modified based on the main 2 cons identified in your comment: the Related Work and the explanation of Graph Query Embeddings, stating clearly the differences. Additionally, we also address your questions one by one.**
>
> **Related Work**
>
> *Poor related work section: Most of the works cited are rather old (e.g. in KGR with logic rules the most recent is 2018, which in machine learning terms is old), there is no comparison of the method proposed and the baselines used, and classic KGR methods that answer queries of type (h,r,?) and complex queries should be clearly distinguished. The related work fails to explain the relation of this work to prior work in a meaningful way.*
>
> Following your recommendation, we have modified and updated the Related Work Section in the new version of the paper. Due to the limit on the comment length, we cannot paste it in this comment, but we recommend checking it in the PDF. As a summary, we have updated part of the bibliography to include mode update works, we have directly stated the difference between this task and Knowledge Completion of Basic Query-Answering. In addition, we have also stated the difference between this work and similar previous ones.
>
> -----------------------
>
> **Graph Query Embedding and Models**
>
> *Clarity, the paper should outline in more detail the Graph Query Embedding method that this paper is based on.*
>
> Likewise, we have also modified the introductory paragraph in Section 3 to reinforce the similarities and differences with Graph Query Embeddings. We attach it to this comment:
>
> Both models presented here follow the idea behind Graph Query Embedding -- GQE: Learning to embed the queries into a low dimensional space. Our models differ from it in the point that logical query operations are represented by geometric operators. In our case, we do not follow the direct geometric sense and these operators are all represented by Neural Networks, instead of just the Intersection operator in GQE. Similarly, however, the operators are jointly optimized with the node embeddings to find the optimal representation.
>
> In order to answer a query, the system receives a query $q$, represented as a DAG, where the nodes are the entities and the edges the relationships. Starting with the embeddings $e_{v_1}, ... e_{v_n}$ of its anchor nodes and apply the logical operations represented by the edges to finally obtain an embedding $q$ of the query.
>
> -----------------------
>
> **Addressing your questions**
>
> *1. Given more than 2 inputs to the intersection operator, how is the order of recursion decided? A different order will yield a different answer.*
>
> The intersection operation with more than 2 input entities indeed selects them recursively. Currently, it is selected following the order of the query. Since the solution is approximate, the results are likely to vary depending on this selection. The reach of this selection is yet to be studied.
>
> *2. What do the colour of the nodes mean in Figure 1?*
>
> We have modified Figure 1 to provide a better explanation. The colors refer to the connection of the edges and the relations in the query.
>
> *3. How is the subgraph (e.g. Figure 1 (C)) extracted from the Knowledge graph?*
>
> This subgraph is never actually extracted, but it was a theoretical explanation of how correct and incorrect answers are extracted in the general Knowledge Graph. We understand this subgraph was giving an incorrect idea of the system, and we hope the new version of the Figure 1 gives a better understanding
>
> *4. How are the initial embeddings s_i for each entity and relation learned?*
>
> The vector space is represented by the entity embeddings. These entity embeddings are jointly learned with the logical operators -- Neural Network weights --. To do so, we use the backpropagation algorithm commonly used to train Neural Networks and the objective function described in Section 3.4. The training objective is to minimize the distance between the query and the query vector, while maximizing the distance from the query to incorrect random entities, via negative sampling
>
> *5. How is the training objective dealing with several possible answers? (I think I know, but the main text should state it clearly)*
>
> The training objective produces one final embedding, the sink node of the Computation Graph. We refer to this embedding as the Query Embedding Representation or the Answer Embedding. Therefore, the loss function only considers one answer embedding. It is only then when all entities are ranked according to the distance value of this embedding to all entity embeddings via near-neighbor search.

---

### Official Review · Reviewer_ukvD · 2021-11-02

**Correctness:** 2
**Technical Novelty And Significance:** 2
**Empirical Novelty And Significance:** 2
**Recommendation:** 6
**Confidence:** 4

**Main Review:**

The paper describes a new approach for learning and inference in knowledge graphs when the queries have logical structure.
The main idea is to use neural networks to implement different operators to compute the queries over the knowledge graph. In particular, mixer-MLP networks have been used to implement these operations. The experiments using standard datasets show performance improvements over state of the art methods.
The main weakness here was that I found it hard to understand the key ideas in the paper. For instance, I am not sure what is being trained differently for the network to “learn” the logical operations. Sections 3.3 seems to indicate that the training is quite standard for MLP-mixer architectures. So essentially, I found it hard to understand why the results were better than the state of the art. I think a bit more of analysis instead of a brief description (3.3, 3.4) will help strengthen the claims of the paper. Right now the details seemed a little fuzzy to me. Also, the improved results in the 2-Vector Average Approach makes it hard to know if the performance improvements were a result of the embeddings actually learning the logical operations since using additional embeddings seems to be helping improve performance. I think the experiment section also needs to analyze the “why” questions more deeply than is being done currently. Overall, the paper shows good results but is perhaps a bit weak on justification for these good results.

After Author review
The authors did a good job in addressing some clarity concerns and some deeper insights I am raising the score accordingly.


**Summary Of The Paper:**

New model for knowledge graph reasoning over complex logical queries.
Shows performance improvements over state of the art methods.


**Summary Of The Review:**

Overall this paper needs a bit more deep analysis and rewriting to back up the experiment results before it can be accepted.

---

> ### Author Response · Authors · 2021-11-22
> **This comment addresses the paper's key ideas and adds the Analysis of Results, which has also been included in the new version of the paper.**
>
> **Models' main idea**
>
> *Understanding the key ideas of the paper*
>
> Let us clarify some of your concerns on the main idea of the models developed in this paper:
>
> The main idea of the architecture is to use neural networks to model the logical operators, regardless of the operator type. It has been proved that neural networks can approximate any continuous function, as such we tried to make most of it and let the model to approximate the patterns of logical operators from the data. It turned out that we do not need a model with strong inductive biases such as query2box to obtain satisfying results on the benchmark. In fact, we are able to obtain better results with a very simple model (i.e., MLP). Our work is in a sense analogous to fuzzy neural networks without hard-constraints.
> The main difference with previous approaches is the fact that we completely disregard the geometrical implications when developing the logical operators, unlike Graph Query Embeddings or Query2Box. We therefore develop all logical operators with Neural Networks and, like the works referred, we jointly learn the parameters of the Neural Networks and the Embeddings with the training objective described in Equation 5 (Section 3.4).
> Following your advice, we have modified Section 3 (Models) and Section 3.1 (Formal Definition) that we hope can help understanding the models implementations. We will try to further explain the adaptation of MLP-Mixer to add on clarification.
>
> ---------------------------------
>
> **Analysis of Results**
>
> *I think the experiment section also needs to analyze the “why” questions more deeply than is being done currently. Overall, the paper shows good results but is perhaps a bit weak on justification for these good results.*
>
> Additionally, we have also included a subsection for the analysis of results. I attach this analysis below for you to read:
>
> In general terms, we observe an increased performance over the selected baselines. In this section, we discuss some implications:
>
> - *Implementation of Logical Operators*. One of the main differences with other approaches such as GQE or Q2B is the use of Neural Networks to represent all logical operations: Projection, Intersection and Negation. In comparison to GQE that only uses it on the Intersection, or Q2B that creates a set of geometrical operations to represent the logical operations. In light of these results, it seems that the geometrical implications of the operations can be restraining some possible solutions.
>
> - *Correctly learning the logical operations*. It is hard to clearly find out if the operators are learning the logical operations correctly. At least, we can assure the Neural Networks do a better job at learning the approximate solution. Neural Networks that implement directly a logical operation are already currently under research. A too constrained implementation of a Neural Logic Network can be found in Appendix D
>
> - *Performance of model variants*. We observe that both Attention Mechanism and the 2-Vector Approach manage to improve the results from the original MLP model. This indicates that Hyper-Graph Embeddings are not correctly transforming the embedding space in our case. Additionally, the improvement original from the 2-vector approach seems to indicate the optimal solution was not yet reached and there is still room for improvement on learning the correct logical operations.

---

### Official Review · Reviewer_rkaY · 2021-11-03

**Correctness:** 3
**Technical Novelty And Significance:** 2
**Empirical Novelty And Significance:** 2
**Recommendation:** 5
**Confidence:** 3

**Main Review:**

In the introduction the description of the figure 1 could be improved. The understanding of what is depicted in the figure is not clear. From the introduction is not clear the embedding process of the queries and the entities.

In section 3 it is not clear how the input embedding for the entities are computed. Even the adoption of the MLP-Mixer in section 3.3 should be better explained.

In section 3.4 it is not clear how the answer vector are computed.

It is not clear the approach used to transform a FOL query to an embedding. Furthermore, how the graph is navigated in order to answer the query is not discussed.

The obtained results seems to be promising when compared to other approaches. However, a discussion of the results should be included in order to explain the motivation of the improvements.




**Summary Of The Paper:**

The paper proposes a new embedding method for knowledge reasoning on knowledge graphs.


**Summary Of The Review:**

Concluding, while the results seem to be interesting the clarity of the proposed approach should be improved.

---

> ### Author Response · Authors · 2021-11-22
> **We have addressed your concerns and modified the paper consequently. In this comment, you can find answers to clarify Figure 1 -- modified in the new version of the paper --, the Embedding Entities, Answer Vector, the Computation Graph and a new Analysis of Results.**
>
> **Figure 1**
>
> *In the introduction the description of the figure 1 could be improved. The understanding of what is depicted in the figure is not clear. From the introduction is not clear the embedding process of the queries and the entities.*
>
> We have modified Figure 1 on the paper to give a better understanding of the system.
>
> ----------------------------
>
> **Embeddings**
>
> *In section 3 it is not clear how the input embedding for the entities are computed. Even the adoption of the MLP-Mixer in section 3.3 should be better explained.*
>
> Regarding the embeddings, we want to clarify that the vector space is formed by the entity embeddings (#entities x entities_dim). These parameters are initialized to zero and learned through back-propagation along with the logical operators -- the Neural Network weights --. The loss function, described in Function 1, aims to do this by minimizing the distance between the query and the query vector, while maximizing the distance from the query to incorrect via negative sampling.
>
> ----------------------------
>
> **Answer Embedding (or Query Vector Representation)**
>
> *In section 3.4 it is not clear how the answer vector are computed.*
>
> *It is not clear the approach used to transform a FOL query to an embedding. Furthermore, how the graph is navigated in order to answer the query is not discussed.*
>
> To infer the answer, the query, provided in a recursive form as a DAG: ((E1, (R1)), (E2, (R2, R3))), is navigated from the anchor nodes to the only sink node, being the edges the relations or logical operations. At the end of this process, an embedding -- the sink node -- in the entity space is obtained, what we call the answer embedding or query embedding representation. Then, all entities are ranked according to the distance value of this embedding to all entity embeddings via near-neighbor search in constant time using Locality Sensitivity Hashing.
>
> ----------------------------
>
> **Computation Graph**
>
> *It is not clear the approach used to transform a FOL query to an embedding. Furthermore, how the graph is navigated in order to answer the query is not discussed.*
>
> We have included the following paragraph to the article in Section 3.1 to address your concerns on the computation graph and its relation to FOL queries:
>
> The Computation Graph can be defined as the Direct Acyclic Graph (DAG) where the nodes correspond to embeddings and the edges represent the logical operations. The computation graph can be derived from a query by representing the relations as projections, intersections as merges and negation as complement. This graph shows the order of operations to answer the queries. Each branch can be computed independently and then merged until the sink node is reached. Each node represents a point in the embedding space and each edge represents a logical operation, computed via a Neural Network in our case. The representation of a FOL as a computation graph can be seen as a heterogeneous tree where each leaf node corresponds to the anchor entities and the root is the final target variable, which is a set of entities.
>
> ----------------------------
>
> **Analysis of Results**
>
> *The obtained results seems to be promising when compared to other approaches. However, a discussion of the results should be included in order to explain the motivation of the improvements.*
>
> We have also included an Analysis of Results, which I attach to this comment:
>
> In general terms, we observe an increased performance over the selected baselines. In this section, we discuss some implications:
>
> - Implementation of Logical Operators. One of the main differences with other approaches such as GQE  or Q2B is the use of Neural Networks to represent all logical operations: Projection, Intersection and Negation. In comparison to GQE that only uses it on the Intersection, or Q2B that creates a set of geometrical operations to represent the logical operations. In light of these results, it seems that the geometrical implications of the operations can be restraining some possible solutions.
>
> - Correctly learning the logical operations. It is hard to clearly find out if the operators are learning the logical operations correctly. At least, we can assure the Neural Networks do a better job at learning the approximate solution. Neural Networks that implement directly a logical operation are already currently under research Neural Logic Networks. A too constrained implementation of a Neural Logic Network can be found in Appendix D.
>
> - Performance of model variants. We observe that both Attention Mechanism and the 2-Vector Approach manage to improve the results from the original MLP model. This indicates that Hyper-Graph Embeddings are not correctly transforming the embedding space in our case. Additionally, the improvement original from the 2-vector approach seems to indicate the optimal solution was not yet reached and there is still room for improvement on learning the correct logical operations.

---

### Decision · Program_Chairs · 2022-01-20

**Decision:**

Accept (Poster)

**Comment:**

This paper focuses on answering complex logical queries over an incomplete KG and use neural networks to do so flexibly handling multiple operations from FOL. Overall reviews agree that empirical performance is impressive. One reviewer gave a strong accept, one leaning to accept and two leaning to reject. Overall, the reviewers who are leaning to reject had mostly clarity issues which seem to have been addressed by the authors (without response from reviewers).
Given this I recomment acceptance.